# Improving Society and the Planet: Sustainability and Fashion Post-Pandemic

**Jessica Strübel [1,\*], Saheli Goswami [1], Ji Hye Kang [1] and Rosemary Leger [2]**

[1] Department of Textiles, Fashion Merchandising & Design, College of Business, University of Rhode Island, Kingston, RI 02881, USA; sgoswami@uri.edu (S.G.); jhkang@uri.edu (J.H.K.)
[2] Marketing, College of Business, University of Rhode Island, Kingston, RI 02881, USA; rosemary_leger@uri.edu
[\*] Correspondence: jessica-strubel@uri.edu

**Abstract:** The COVID-19 pandemic has exposed the vulnerability of our ecosystem and demonstrated a crucial need to address sustainability across the fashion supply chain, including the end-use consumer. As consumers become more conscious and demand sustainable fashion, the question of whether the pandemic has shaped such behaviors for long-term transitions remains unanswered. This qualitative study aimed to understand whether the COVID-19 pandemic has created a societal shift in individual attitudes toward sustainable fashion, whether it can help to motivate long-term sustainable consumption practices, and whether positive psychology plays a role in this context. With online survey data collected from 154 US consumers, summative content analysis and thematic analysis results revealed that as consumers continued to be puzzled by what constitutes sustainability, their pro-sustainability attitudes and behaviors did not evolve as claimed by prior literature. However, not only positive emotions associated with care and concern motivated consumers' pro-sustainability, but indeed post-purchase positive feelings of contentment and altruism were revealed, affirming the need for marketing messages to incorporate positive psychology perspectives to motivate long-term sustainable fashion consumption practices.

**Keywords:** sustainability; fashion; COVID-19 pandemic; consumers; sustainable consumption; positive psychology

## 1. Introduction

The COVID-19 pandemic has changed the fashion industry's landscape, with unprecedented challenges, emphasizing the humanitarian and environmental concerns of this industry [1]. As fashion brands experienced a massive drop in their revenues during the multiple rounds of global lockdown, employees were furloughed and terminated from practically all job positions [2]. The pandemic affected the global fashion supply chains; canceled orders with suppliers led to the extensive criticism of brands while causing their large, unsold inventories to rapidly go out of fashion. Critics acknowledge the pandemic as a driving moment for fashion brands to shift to greener and ethically responsible approaches in their future business practices.

Interestingly, the literature presents a rather contrasting picture when it comes to consumer behaviors and COVID-19's impact on them. For example, some studies claim that the pandemic drastically impacted consumer behaviors [3]. Lockdowns led consumers to shift from traditional branded fashion products to domestic consumption, such as resorting to repairs and do-it-yourself projects [4,5], and consumers increasingly preferred sustainable alternatives and a circular economy when shopping for fashion items [6]. However, other studies claim that consumers' price sensitivity persisted [3]. Fast fashion brands' (such as Shein) success during COVID-19, despite their unsustainable practices [7,8], have raised doubts about the pandemic's influence on consumers' behaviors, particularly their sustainable choices. Similarly, consumers' preferences for sustainable fashion continue to

lack accountability allegedly, and their continued struggle to identify or access sustainable products causes them to still demand that policymakers tackle unsustainability [3]. Thus, while, on the one hand, the pandemic has been reported to trigger a shift in fashion consumers in favor of sustainability, on the other hand, it is argued that the pandemic merely reintroduced sustainability as a buzzword in the fashion industry [9], with the majority of its consumers still searching for affordable and easily available fast fashion items. Such conflicting research leaves uncertainty about whether the pandemic truly prompted a sustainable fashion shift, forming the first research gap of this study.

Additionally, the advent of COVID-19 heralded numerous unprecedented economic and social shifts in consumers' lives. It created a landscape of chaotic and far-reaching changes in consumer lives, including prolonged financial instability instigated by job losses [10], a blurred work–life balance with remote working enforcements, and restricted social interactions with social distancing, but also the development of new relational goals with increased family time [11]. Given the profound influences that COVID-19 exerted on consumers' psychological value orientations and life objectives [12], it is important to contemplate whether the pandemic could support a more long-term transition in consumers' fashion behaviors and embed sustainability. In other words, if indeed COVID-19 prompted consumers to adopt more sustainable approaches to fashion consumption, can the same pandemic help to sustain such changes for the long term? This question gains significance, as many scholars and experts tout the pandemic as the start of a new era in sustainable fashion [13], accompanied by fashion brands transitioning toward more sustainable business practices. Consequently, it warrants our attention and represents the second research gap addressed by this study.

Furthermore, in response to consumers' argued preferences for sustainable alternatives [6], many fashion brands have started to incorporate sustainability into their business models [14]. However, what is unknown is what exactly is the most effective way to shift consumer behavior to be more sustainable. Some sources suggest that consumers follow the actions of others, while others suggest that a more successful strategy is to apply penalties for bad behavior [15]. There is also evidence to suggest that positive psychological messages in marketing and media can create more long-term value shifts in consumers with respect to sustainability and consumerism [16]. Although considerable research efforts have been directed at understanding sustainable consumption, contradictions in the existing literature further support the need for the current study.

Thus, to address the research gaps, this study aimed to explore whether COVID-19 influenced consumers' sustainable fashion consumption practices and whether the pandemic can be leveraged to transition to long-term sustainable behaviors among fashion consumers. A qualitative inquiry was utilized because of its exploratory approach. Specifically, responses to 15 open-ended survey questions provided useful insights into the attitudes, beliefs, and affective responses of consumers in relation to sustainable consumption. This study contributes to the discussion of fashion sustainability in the aftermath of COVID-19 but from consumers' perspectives. While a considerable number of prior studies have reported contradictory findings regarding consumers' preferences for sustainable fashion, as shared earlier, this research brings much-needed clarity on what consumers really think and how they really behave when it comes to sustainable fashion consumption as a result of the pandemic. Furthermore, this study also sheds light on how COVID-19 can create and motivate long-term consumer preferences for sustainable fashion and whether positive psychology plays a role in the same. Accordingly, this study makes a theoretical contribution to the sustainable fashion and COVID-19 literature.

## 2. Literature Review

### 2.1. Sustainability in Fashion Industry and the COVID-19 Pandemic

Fashion and apparel are multi-trillion-dollar global industries [17]. While these industries support many economies, employing approximately 86 million people (mostly women) around the world, they have also been associated with significant social and

environmental impacts, such as the unfair treatment of laborers, low pay, poor working conditions, the production of 8–10% of world's greenhouse gas emissions, increased ocean plastic pollution, low recycling participation (leading to the loss of USD 500 billion of resources), and a general lack of transparency that has created skepticism and decreased trust among consumers [18–20]. Such a wide array of challenges within the fashion industry, associated with its impacts on both people and the planet, have created heightened interest in the industry's sustainability impacts among international organizations [21,22], NGOs [23], and scholarly researchers. Particularly in the last few decades, researchers in fashion marketing, consumer behavior, and management areas have repeatedly called the industry's attention to an increased need for sustainable practices applied to the social, environmental, and economic realms of the industry [24–28].

Such an awareness of sustainability within the industry was noted to varying extents in the pre-COVID-19 years. Several signs and examples suggested that sustainability was arguably one of the biggest buzzwords within the fashion industry, with brands beginning to implement a wide range of sustainability initiatives in their businesses. For example, some of the world's largest fashion brands, including H&M, Gap, Everlane, Levis, and Stella McCartney, increasingly jumped on the sustainability bandwagon [9]. Brands started advertising and announcing their endorsement of sustainable practices, ranging from recycling brands' post-consumer wastes [29,30], reducing water pollution and carbon footprints [31], supporting fair trade practices and implementing transparency [32], to giving back to the community [33]. Such initiatives were not only incentivized by fashion consumers' frequently cited intentions of supporting sustainable fashion brands [31,34] but also by the United Nations' Sustainable Development Goals, announced in 2015, the Fashion Industry Charter for Climate Action from 2018, as well as the United Nations' (UN) dedicated Alliance for Sustainable Fashion, developed in 2019 [9,17]. These examples and more indicate that both consumers and the fashion brands were perhaps hopeful about sustainability gaining enough traction and momentum to manage the industry's social and environmental footprints.

However, with COVID-19 bringing the entire fashion industry to a halt during the early months of 2020, the fashion industry has been hit hard since then. As the worldwide demand for apparel, shoes, and fashion accessories plummeted, with fashion products being perceived as "non-essential" goods by consumers, unsold inventories placed fashion brands in a state of financial distress [35]. Furthermore, as fashion weeks and trade shows were canceled, stores were shut down, and supply chains were disrupted, the industry's global manufacturing networks underwent a devastating impact; fashion brands were forced to renege on payments for bulk orders already manufactured, and suppliers were forced to face losses worth billions [36,37]. The growing financial crisis among furloughed employees and unpaid workers left the fashion industry in a precarious situation [38] that further worsened the impending humanitarian crisis of the fashion industry from its pre-pandemic era. Thus, while, on the one hand, fashion businesses, including retailers, brands, and suppliers, were exposed to major disruptions and devastating impacts brought about by COVID-19, on the other hand, consumers suffered financial anxieties and job insecurities [39].

Interestingly, a significant amount of research has investigated the impact of the pandemic on fashion sustainability, primarily from brands' and businesses' perspectives. For example, studies have investigated how many fashion brands suffered the perils of unsold inventories [40] and how small fashion businesses struggled during the pandemic [41,42]. Scholars have questioned whether the pandemic could support the fashion industry in its transition to sustainable practices [9]. Yet other scholars have indicated how the pandemic further amplified the fundamental lack of social sustainability in the fashion industry and remarked on the need for the supply chain to strategize for an appropriate redressal [43]. While these and other studies on the impact of COVID-19 on the fashion industry from businesses' perspectives remain eminent, such studies fail to paint a picture from consumers' perspectives. In particular, with a few notable exceptions [39,44,45], currently, an

exhaustive study is lacking to document and determine if and how fashion consumers' sustainable consumption practices have evolved due to the pandemic.

### 2.2. Consumer Sustainability-Related Behavior and COVID-19 Pandemic

As mentioned earlier, the pandemic has had a profound impact on various aspects of people's lives, triggering changes in their behaviors and perceptions of the external world, and, subsequently, alterations in consumption patterns and marketplace adaptation. Initially, as a response to the uncertainty and fear surrounding the pandemic, some consumers actively resorted to hoarding and experienced a surge in the consumption of essential items such as toilet paper and hand sanitizer [46,47]. Furthermore, the pandemic has led to a transformation in consumers' perspectives and the significance that they attribute to products and services [48]. In particular, consumers have developed a strong inclination toward value-oriented consumption [49,50]. For instance, consumers have shown a tendency to exhibit loyalty toward products that align with their ideal self-concept, which helps to alleviate perceived risks and fears [49]. By choosing products that resonate with their values and aspirations, consumers feel a sense of connection and authenticity [51]. This loyalty extends to well-known and reputable brands that convey trust, as they provide consumers with a sense of security amidst uncertainty [46]. These shifts in consumption patterns and reevaluation have also manifested in the realm of fashion.

The heightened awareness of humanitarian and environmental concerns during the pandemic has led consumers to develop a greater interest in a sustainable economy. On the one hand, consumers have shown stronger social identifications and communal feelings, leading them to support local businesses, which aligns with the concept of social sustainability [52]. Additionally, this awareness of sustainability has led consumers to become more conscious shoppers, reassessing their purchasing decisions around nondurable goods such as clothing [53–55]. Interestingly, the symbolic meaning of sustainable fashion and the coping mechanism of fear have been found to motivate the consumption of sustainable brands during the pandemic. Consumers who feel threatened by the pandemic are more inclined to engage in materialism as a means of compensating for their diminished sense of power [56,57]. Sustainable fashion brands, in this context, are viewed as a new status symbol, attracting consumers who experienced higher levels of fear during the pandemic [58].

Nevertheless, despite the growing awareness of sustainable fashion, purchasing behavior in the fashion market remains diverse. Some consumers prioritize obtaining more value for their money, particularly in light of the ongoing pandemic [19,54]. The ability to effectively manage risks and crises through consumer behavior is heavily influenced by wealth distribution [59]. Therefore, consumers who faced financial instability during the pandemic may have limited spending capacity and resources to support sustainable initiatives [60]. This has led to the remarkable growth of lower-priced fast fashion brands such as Shein [8]. Simultaneously, the emergence of thrifting as a new habit during the pandemic has played a significant role in fostering the growth of the resale market [61]. These trends reflect the value placed on sustainability alongside economic constraints, evoking positive emotions such as fun, excitement, and pride in second-hand fashion products [61,62]. Furthermore, an industry report indicated that 76% of first-time second-hand clothing buyers in 2020 intended to increase their spending on second-hand clothing in the next five years [62], suggesting that the adoption of sustainable fashion consumption and the development of new habits of thrifting are likely to persist beyond the pandemic.

Despite the varied responses to sustainable fashion during the pandemic, as discussed earlier, there is a notable absence of comprehensive research documenting and examining the evolution of fashion consumers' sustainable consumption practices in response to the pandemic. Therefore, it is crucial to conduct studies to understand whether the pandemic has triggered a societal shift among consumers toward sustainability and has influenced attitudes regarding fashion sustainability on a broader scale.

Moreover, when comparing the impact of the pandemic to significant social threats such as the Great Depression, researchers have put forth arguments suggesting that the psychological effects of the pandemic may have enduring consequences [46]. However, the existing literature on the long-term effects of social threats has yielded mixed findings. For instance, Phipps and Ozanne (2017) [63] demonstrated that individuals who experienced disruptions to their norms, beliefs, routines, and practices during a natural disaster developed new habits that persisted even after the threat had subsided. Conversely, other studies propose that people tend to adopt a fresh start mindset, creating a clear demarcation between the period before and the period after the social threat [64].

In the context of sustainable fashion consumption, it is crucial to further investigate the sustainability practices that emerged during the pandemic, if any, and their potential longevity. Understanding the sustainability-related behaviors that have been influenced by the pandemic can provide valuable insights into the future trajectory of sustainable fashion consumption and its long-term societal impact.

### 2.3. Positive Psychology and Media Messages

Psychologists believe that theory and psychology can be used to mitigate sustainability challenges, such as climate change [65,66]. More specifically, there is a need to understand how to change consumer awareness, attitudes, and practices consistent with the needs of struggling economies and a suffering planet. Research by consumer sociologists has been examining strategies to increase sustainable consumerism [67]. Due to the complexity of feelings and emotions, research has produced contradicting findings. Some research suggests that while negative emotions such as guilt can prompt environmentally friendly actions, guilt-inducing advertisements do not necessarily motivate consumers to consume sustainably [67]. Other research suggests that while the majority of sustainability practices are devised from a primary prevention perspective, the use of reflexivity, emotional intelligence, and interpersonal sensitivity can help to foster an altruistic approach among people and organizations to help them to focus on the promotion of mutual gains and collective sustainable development [68,69].

Fashion brands can encourage sustainable consumerism and environmentally responsible choices through positive social cues for sustainability in their marketing. Although negative signals may condition people to modify their behavior, research has shown that positive cues and emotions, such as feelings of joy, pride, and social connection, can motivate people to engage in sustainable behaviors [16,70]. For example, consumers who feel a sense of pride and accomplishment from recycling or reducing their carbon footprint are more likely to continue engaging in those behaviors.

In addition, positive psychology messages can also promote a sense of meaning and purpose in consumers, which can lead to a shift away from non-sustainable consumerism. For example, messaging that emphasizes the importance of sustainability and solutions to protect the environment for future generations can help consumers to see the long-term benefits of sustainability and feel a sense of responsibility toward the planet.

### 2.4. Purpose Statement

We seek to develop an understanding of the role of positively motivated decision-making for sustainable consumer behavior in fashion and apparel products and how to change fear-driven determinants to positive motives, attitudes, and sustainable practices in the wake of COVID-19. Ultimately, we want to offer comprehensive and effective strategies to guide both fashion consumers and brands to transition toward sustainability. Therefore, the purpose of the project was to (1) understand how long-term sustainable fashion consumer behavior changed due to the pandemic and (2) explore how marketing can effectively educate consumers toward long-term sustainable consumption behavior. The following specific research questions were developed to achieve the purpose of the project.

RQ1: How has the pandemic triggered a societal shift in consumers in favor of sustainability and a change in attitude es about fashion sustainability at a societal level?

RQ2: Can the pandemic be used to create long-term value shifts in consumers for sustainability?

RQ3: Can positive antecedents and consequences (social and/or psychological) motivate consumers to adopt sustainable consumption practices for the long term?

## 3. Method

### 3.1. Participants

After receiving Institutional Review Board (IRB) approval from the authors' university, participants were recruited to participate in this study. A research panel was recruited directly from Qualtrics (a market research service), and participants each received USD 5.00 in financial compensation.

The final sample for the study consisted of 154 participants. Of the total participants, 75 (48.7%) were men, and 78 (50.6%) were women. Participants ranged in age from 18 to 77 ($M_{age}$ = 36.60 years, SD = 14.93). In terms of ethnicity, the majority were White (73.4%, *n* = 113), fifteen (9.7%) were African American/Black, eleven (7.1%) were Hispanic/Latino/Latinx, nine (5.8%) Asian or Pacific Islander, and six (3.8%) identified as "other" ethnicity.

### 3.2. Data Collection and Analysis

We utilized an online survey composed of 16 open-ended questions to explore the current topic and obtain a thorough understanding of consumer attitudes about sustainability and business practices during the pandemic. Questions addressed participants' views of sustainability, their purchase behavior (past and present) of sustainable fashion products, the impact of the COVID-19 pandemic on their consumer behavior, and the effectiveness of marketing messages upon their consumer behavior. To stay true to the exploratory nature and purpose of this research, no specific examples were adopted connecting sustainability to the fashion industry, to refrain from priming our participants.

Due to the sample size, we utilized summative content analysis for the systematic examination of six questions, such as "Who or what inspired/motivated your purchase of sustainable fashion?". For the content analysis of the six questions, the first author reviewed the responses and constructed coding categories from existing words and phrases. After the fourth author coded the data using the established codes, the codes were counted, and we performed an intercoder reliability check [71].

For the remaining ten questions, such as "How did you feel when you bought and used sustainable fashion?", data were analyzed using thematic analysis, which is a form of inductive analysis. Qualitative induction is characterized by identifying patterns in the data and making generalizations based on the observations. After the initial inspection of the data, the first author read the raw data repeatedly and began identifying initial labels for important features in the data based on disassembling the responses provided by the 154 participants. After a thorough review of the responses during the open coding stage, the first author generated the initial coding categories [72–75]. The codebook continued to develop through the inductive process [72,74] by looking for the presence of attributes and the frequency of occurrence of key words and ideas as we coded the dataset and then identified patterns (i.e., commonalities and differences) to derive themes from the data [76,77].

During the second phase of coding, the themes were checked against the coded data. In this stage, several categories were eliminated, collapsed, and combined into central themes based on the similarities in responses. The broader categories were systematically developed into subcategories. During the third and final phase of coding, the first author reviewed and refined the themes after cross-checking the categories with the terms and phrases used in the sustainability and consumer psychology literature [68,78]. The fourth

author then independently coded the participants' responses using the NVivo software R1 and the final coding scheme.

Data Reliability

The reliability of the coding process was evaluated for this study using Cohen's kappa ($\kappa$) [79]. The inter-coder agreement was calculated using Holsti's Formula [80]. For the summative content analysis, the intercoder agreement ranged from 97.25 to 100%. Kappa values ranged from 0.31 (fair agreement) to 1 (perfect agreement). For the inductive thematic analysis, intercoder agreement ranged from 97.49 to 100%. Kappa values ranged from 0.21 (fair agreement) to 1 (perfect agreement).

## 4. Findings

In this section, we present the detailed analysis of the data and the main themes identified in relation to the research questions.

### 4.1. Defining Sustainability: Consumer Misconceptions

We identified five broad themes in respondents' definitions of sustainability: environmental concerns, mindful of self, reclamation practices, restrictive practices, and social responsibility (see Table 1).

**Table 1.** Definition of sustainability.

| Primary Theme | Frequency, *n* | Subthemes | Description |
|---|---|---|---|
| Environmental Concerns | 64 | Environmentally friendly<br>Local sourcing<br>Natural fibers | Sustainability is defined as concern for minimizing the harmful effects on the environment; Protect planet; Reduce pollution; Reduce carbon footprint. |
| Mindful of Self | 19 | Affordability<br>Comfort | Sustainability is defined as awareness of the present moment in relation to oneself, especially with respect to affordability and comfort. |
| Reclamation Practice | 94 | Longevity<br>Multi-use<br>Reusable/recycle | Sustainability is defined as recovering products/brands from dispossession (e.g., composting, second-hand, thrift, care, recycle, re-use, mending), which implies a long existence or service. |
| Restrictive Practice | 114 | Conserve<br>Efficient<br>Optimization<br>Preserved stability<br>Quality<br>Renewable | Sustainability is defined as limiting one's consumption of particular products or brands and even quantity of items to prevent the wasteful or harmful overuse of (a resource). More specifically, it refers to achieving maximum productivity with minimum wasted effort or expense, making the best or most effective use of a resource that is capable of being renewed while maintaining its original or existing state or general excellence of standard. |
| Social Responsibility | 28 | Beneficial<br>Caring for Community<br>Responsible<br>Safe | Sustainability is defined as care and obligation for the long-term benefits to society, which may include creating a society that is free from danger and risk. |
| Unclear response | 42 | | Participant provided a response that was unrelated to the question. |
| Nothing | 3 | | Participant did not provide a response. |
| Unsure | 2 | | Participant claimed that they did not know how to define it. |

The most popular category was *restrictive practices* (*n* = 114), which can be defined as limiting one's consumption of particular products or brands and even the quantity of items in order to conserve water, electricity, and fuel. This can also manifest as a restriction on plastic usage, waste reduction, and a focus on the quality and longevity of a good. Participants talked about preventing waste and the overuse of resources and maintaining the current condition of the planet. There was no significant gender or age difference among the participants. Within this category, participants talked predominantly about preserving the planet in its current state and conserving resources—for example, "not running the planet by using natural resources", "to be able to continue something forever", and "the ability to maintain something".

*Reclamation Practice* (*n* = 94) was the second most popular theme found in the data. Reclamation refers to recovering products through re-use, mending, and recycling. Participants also referred to the longevity of products and services over a period of time as a result of higher quality and maintenance: "It means something that is long lasting and can continue for many years after its inception".

It is worth noting that 44 participants provided ambiguous responses or claimed that they had no answer to our question. These findings demonstrate that despite the large body of literature on sustainability, there are significant levels of confusion and knowledge gaps with respect to what constitutes sustainability.

### 4.2. Defining Sustainable Fashion

For defining sustainable fashion, we classified participants' definitions of sustainable fashion utilizing the same five themes as above (see Table 2). For sustainable fashion, however, *reclamation practice* (*n* = 90) was the most popular thematic category, followed by environmental concerns (*n* = 73; minimizing the harmful effects on the environment). Participants commonly defined sustainable fashion as second-hand or fabricated from recycled material, demonstrating that recycling is a high priority.

"Something that can be recycled or is made from something recycled".

"Clothes made ethically from renewable or recycled environmentally friendly resources".

"They are products that do not contribute to plastic waste. They are made from renewable materials".

For participants who focused more on environmental concerns, they referenced items that did not cause harm to the environment because they were made of "natural" materials, were "animal friendly" or locally sourced, or didn't produce by-products that would damage the environment:

"Sustainable fashion to me is apparel, shoes, jewelry, accessories, etc., being eco friendly made and how they are made doesn't have a negative effect on the environment".

"Probably items made out of non-animal sources (fabric vs. leather, etc.) But this can be problematic because some fabrics are not environmentally friendly. For example, cotton crops use huge amounts of water. California produces a lot of cotton. California is in a severe drought".

"Sustainable fashion is accessories from ingredients that decompose safely without harming wildlife".

As with the previous question, a significant number of participants provided no answer or unclear responses (*n* = 37) to the question. Because there is no clear universal definition of what constitutes sustainable fashion, this is to be expected.

**Table 2.** Definition of sustainable fashion (N = 153).

| Primary Theme | Frequency, *n* | Subthemes | Description |
|---|---|---|---|
| Environmental Concerns | 73 | Environmentally friendly<br>Animal Friendly Products<br>Local sourcing<br>Natural fibers | Sustainability is defined as concern for minimizing the harmful effects on the environment; Protect planet; Reduce pollution; Reduce carbon footprint |
| Mindful of Self | 27 | Affordability<br>Comfort<br>Design | Sustainability is defined as awareness on the present moment in relation to oneself, especially with respect to affordability and comfort. |
| Reclamation Practice | 90 | Longevity<br>Multi-use<br>Reusable-recycle | Sustainability is defined as recovering products/brands from dispossession (e.g., composting, second-hand, thrift, care, recycle, re-use, mending), which implies a long existence or service. |
| Restrictive Practice | 21 | Conserve<br>Preservation<br>High quality<br>Renewable | Sustainability is defined as limiting one's consumption of particular products or brands and even quantity of items to prevent the wasteful or harmful overuse of (a resource). More specifically, it refers to achieving maximum productivity with minimum wasted effort or expense, making the best or most effective use of a resource that is capable of being renewed while maintaining its original or existing state or general excellence of standard. |
| Social Responsibility | 12 | Beneficial<br>Caring for Community<br>Beneficial<br>Responsible<br>Safe | Sustainability is defined as care and obligation for the long-term benefits to society, which may include creating a society that is free from danger and risk. |

*4.3. Consumers' Pro-Sustainability Behaviors*

We also asked participants a series of questions about their own sustainable fashion purchases pre- and post-pandemic and how the COVID-19 pandemic impacted their sustainable fashion consumption (see Table 3). The majority (*n* = 68) said that the pandemic had no effect on their purchase of sustainable fashion goods, 30 participants said that they purchased fewer fashion items in general, and 13 participants said that the pandemic had raised their individual awareness about sustainability. There did not appear to be any significant gender or age differences in the responses.

In our sample, 101 respondents purchased sustainable fashion products prior to the pandemic, while 51 did not. Again, there were no significant gender or age differences in the responses. After the pandemic, 97 claimed to have purchased sustainable fashion products, and 57 did not (no significant gender or age differences).

When asked about what types of sustainable fashion products were purchased pre-pandemic, the most popular response was tops, followed by shoes and accessories (e.g., jewelry and handbags; see Table 4). For those individuals who did purchase sustainable fashion products post-pandemic, they once again purchased tops, shoes, and accessories. Post-pandemic, women purchased sustainable bottoms (including jeans) at the same rate as shoes (*n* = 17).

**Table 3.** Impact of COVID-19 on purchases (N = 153).

| Theme | Frequency, *n* | Description |
|---|---|---|
| Quality | 2 | Due to COVID-19, participant purchased better-quality items. |
| Accessibility | 1 | Clothing is/was less accessible during peak of COVID-19. |
| Less concerned | 1 | Less concerned with appearance due to social distancing and staying at home. |
| More awareness | 13 | More awareness of sustainability during COVID-19. |
| More recycling | 2 | Participate in recycling clothing; second-hand purchases. |
| Comfort | 1 | Purchased more clothing for comfort. |
| No change | 68 | Sustainable consumer behavior remained stable during and after COVID-19. |
| Not applicable | 10 | This question does not pertain to the participant (per the respondent). |
| Purchased less | 30 | Purchased fewer sustainable items and/or fewer items in general. |
| Purchased more | 8 | Purchased more sustainable items and/or more items in general. |
| Unclear/other response | 17 | Participant provided a response that was unrelated to the question. |

**Table 4.** Product consumption (N = 153).

| Primary Product Categories | Pre-Pandemic Frequency, *n* | Post-Pandemic Frequency, *n* |
|---|---|---|
| Accessories | | |
|   Jewelry | 15 | 10 |
|   Bracelet | 3 | 4 |
|   Necklace | 0 | 3 |
|   Ring | 1 | 0 |
|   Handbags | 10 | 7 |
|   General | 5 | 4 |
| Bottoms | | |
|   All types | 11 | 17 |
|   Jeans | 9 | 10 |
| Brands | | |
|   General branded items * | 4 | 7 |
|   H&M | 1 | 1 |
|   Hollister | 1 | 1 |
|   Sorona | 0 | 1 |
| Intimates | 4 | 2 |
| Makeup/Fragrance | 3 | 1 |
| Outerwear | 4 | 3 |
| Shoes | | |
|   General ** | 34 | 32 |
|   Sandals | 0 | 2 |
|   Sneakers | 3 | 1 |
| Suiting | 1 | 2 |
| Tops | 48 | 49 |
| No answer | 54 | 58 |
| Unclear response | 4 | 3 |

* No specific brand was mentioned. ** No specific style of shoe was mentioned.

*4.4. Positive Social and Psychological Signals*

4.4.1. Barriers

For nearly one third of participants who said that they did not purchase any sustainable fashion product pre-pandemic, the majority (*n* = 16) did not provide a clear reason (see

Table 5). The next most popular reason for not purchasing sustainable fashion (*n* = 13) was that it is too costly, followed by a self-acknowledged lack of awareness (*n* = 11).

"I sometimes think it's overrated and too much money".

"I am broke and can't afford it".

"I honestly never thought about it before I wasn't aware it was a thing".

"It just wasn't something I knew much about".

**Table 5.** Barriers to sustainable fashion purchases (N = 153).

| Primary Theme | Description | Pre-Pandemic Frequency, *n* | Post-Pandemic Frequency, *n* |
|---|---|---|---|
| Accessibility/availability | Lack of access to sustainable fashion. | 3 | 4 |
| Too costly | Sustainable fashion costs too much. | 13 | 21 |
| Dislike of design | Participant claimed to not like the design and/or color of sustainable fashion products, or they did not consider it trendy. | 3 | 3 |
| No need | Participant felt no need for sustainable fashion. | 8 | 12 |
| Lack of trust | Lack of trust in the brands claiming a product is sustainable. | 0 | 1 |
| Lack of awareness | Unclear on what constitutes sustainable fashion, where to find it, etc. | 11 | 3 |
| Want new products | Participant equated sustainable fashion with second-hand clothing | 1 | 0 |
| No answer | Participant did not provide a response. | 85 | 78 |
| Don't believe it | Participant stated that they did not believe in the concept of sustainable fashion. | 2 | 2 |
| Don't care/not important | Participant said they did not care about practicing sustainable behaviors. | 9 | 13 |
| Unclear response | Participant provided a response that was unrelated to the question. | 16 | 14 |
| Unsure | Participant claimed that they did not know how to define it. | 2 | 2 |

The primary *post*-pandemic barrier to purchasing sustainable fashion was cost (*n* = 21). The COVID-19 pandemic had a significant social and economic effect on society across the globe. Many people lost their jobs, and many people had to reallocate and prioritize their personal finances simply for survival. Something as costly (or perceived to be costly) as sustainable products was not viewed as a necessity despite an increase in interest in the environmental impact of fashion consumption.

Participants also claimed a general lack of concern or apathy (*n* = 13) as a barrier to purchasing sustainable fashion post-pandemic: "[B]ecause i said i didn't want to nor do i want to now, and i do not feel any emotion towards this". When people were overwhelmed with financial concerns during the pandemic, it was to be expected that there would be a degree of apathy toward behaviors that are potentially considered to be frivolous. See Table 5 for a complete list of findings.

### 4.4.2. Motivations

For the respondents who did purchase sustainable fashion products pre- and post-pandemic, there were a variety of motivations to participate in this behavior (see Table 6).

Environmental concerns (*n* = 72) were the primary motivator, followed by product characteristics (*n* = 58; design, durability, fabrication) and a more self-focused reason (*n* = 50). There was no significant difference between men and women, nor between age groups.

"I want to be responsible and do my part on cutting my carbon foot print. So I can leave the world a better place. We all, want cleaner air to breath and cleaner water".

"The look of the shoe and the material motivated me to buy it".

"I want to feel good so I bought the products".

**Table 6.** Motivation to purchase sustainable fashion (N = 153).

| Primary Theme | Subthemes | Description | Pre-Pandemic Frequency, *n* | Post-Pandemic Frequency, *n* |
|---|---|---|---|---|
| Environmental Concern | Climate Change Protect Animals Reduce Pollution Reduce Waste | Motivation to purchase was motivated by a concern for minimizing the harmful effects on the environment; Protect planet; Reduce pollution; Reduce carbon footprint | 72 | 52 |
| Social Concern | | The motivation to purchase sustainable fashion is cited as being a concern and obligation for the long-term benefits to society. | 6 | 3 |
| Cost | | Participant cited the cost as being the primary motivation for purchasing sustainable fashion. | 9 | 17 |
| Media | Advertisements Catalogs Magazines Social Media Youtube | | 24 | 23 |
| Other People | Celebrity Designer-Brand Family Peers Significant Other University Course | Participant cited an external source (i.e., person) as the motivation to purchase sustainable fashion. | 28 | 33 |
| Product | Availability Design Durability-Longevity Fabrication-Ingredients Fit Product Type Quality | Participant described specific characteristics of the fashion product as the motivation for its purchase. | 58 | 40 |
| Self-motivated | Affect Need Work/Social Role | Participant cited an internal source (i.e., self) as the motivation to purchase sustainable fashion, such as their own perceptions, beliefs, feelings and behavioral tendencies towards sustainable behavior. | 50 | 52 |
| No Answer | | Participant did not provide a response. | 101 | 107 |
| Don't Care-Not Important | | Participant said they did not care about practicing sustainable behaviors. | 1 | 0 |
| Nothing-No One | | Participant stated that nothing nor no one motivated them to purchase sustainable fashion. | 3 | 7 |
| Unclear Response | | Participant provided a response that was unrelated to the question. | 4 | 11 |

The motivation to purchase sustainable fashion products did not shift considerably after the pandemic. Environmental concerns ($n = 52$) were still the primary motivator in conjunction with self-focused reasons ($n = 52$). However, post-pandemic, more participants mentioned cost as a motivator ($n = 17$): "Long term money saving and for the planet", "It's often cheaper to thrift and it lessens your pollution footprint". The last statement illustrates a view of sustainability that deviates from others that view sustainable goods outside of their price range. Many people associate sustainability with thrifting and therefore view it as less expensive than organics and "traditional" sustainable items.

### 4.4.3. Post-Purchase Feelings

For those individuals who did make sustainable purchases pre- and post-pandemic, we inquired about how the purchase(s) made them feel (see Table 7). The participants responded with a variety of attitudes, emotions, moods, values, and feelings. For pre-pandemic purchases, the most frequent themes were self-focused feelings of contentment ($n = 29$), joyous emotion ($n = 20$), and feelings of confidence ($n = 19$), such as "It makes me feel good" or "I feel better about myself". There were fewer individuals who indicated that their sustainable purchases were related to altruistic values ($n = 19$).

> "I felt great because I was doing something to help protect the Earth from being damaged anymore than we are already damaged it".

> "I felt like I did something good for the environment and that I contributed to the environment by not using harmful product".

**Table 7.** Feelings about sustainable fashion purchases (N = 153).

| Categories and Subcategories | Description | Pre-Pandemic Frequency, *n* | Post-Pandemic Frequency, *n* |
|---|---|---|---|
| Attitudes | Participant alluded to a relatively stable pattern of beliefs, feelings, and behavioral tendencies toward sustainable fashion. | | |
| Negative | | 1 | 0 |
| neutral/normal | | 2 | 2 |
| positive | | 8 | 9 |
| Emotions | Participant referenced physical states that arise as a response to external stimuli or events that are observable. | | |
| Enthusiasm | | 5 | 5 |
| Fear | | 1 | 0 |
| Joy | | 20 | 3 |
| Feelings | Participant mentioned perceptions of sensation in the body that can be hidden. | | |
| Confident | | 19 | 8 |
| Contentment | | 29 | 23 |
| Happiness | | 7 | 10 |
| Love | | 1 | 1 |
| Pride | | 4 | 4 |
| Worry | | 2 | 0 |
| Positive mood | Participant mentioned a positive state or quality at a particular time. | 1 | 0 |
| Values | Participant alluded to a central higher-order set of preferences for goals in life and ways of living that are felt to be ideal and important. | | |
| Altruism | | 19 | 23 |
| Compassion | | 1 | 0 |
| Honesty | | 1 | 0 |

**Table 7.** *Cont.*

| Categories and Subcategories | Description | Pre-Pandemic Frequency, *n* | Post-Pandemic Frequency, *n* |
|---|---|---|---|
| No answer | Participant did not provide a response. | 56 | 59 |
| Don't believe it | Participant stated that they did not believe in the concept of sustainable fashion. | 0 | 1 |
| Don't care/not important | The participant said they did not care about practicing sustainable behaviors. | 1 | 0 |
| No feelings/indifferent | Participant said they felt nothing, or they had mixed feelings | 3 | 3 |
| Unclear response | Participant provided a response that was unrelated to the question. | 5 | 8 |
| Unsure | Participant claimed that they were uncertain about their feelings. | 0 | 1 |

*Post-pandemic*, feelings of contentment (*n* = 23) and altruistic values (*n* = 23) were the most popular responses, followed by feelings of happiness (*n* = 10).

*4.5. Marketing Messages*

Finally, we asked participants to discuss the types of messages that fashion companies/brands used that encouraged them to behave sustainably (see Table 8). Responses were first coded for message content and message tone. Then, the data were reevaluated to determine subthemes within these two broader categories. Responses overwhelmingly indicated message content as encouraging them to participate in sustainable behaviors (*n* = 102) compared with message tone (*n* = 44).

**Table 8.** Influential sustainability messages (N = 153).

| Primary Message Themes | Message Subthemes | Description | Total Frequency |
|---|---|---|---|
| Message content | | The information contained within communication media. | 102 |
| Concerned actions and practices | Reclamation<br>Rejection<br>Restriction<br>Animal-friendly products<br>Care of product<br>Brand-specific<br>Materials<br>Packaging<br>Product quality<br>Economical concerns<br>Environmental concerns | Display of sustainable behaviors in the messaging that could be mirrored by the consumer. | 34 |
| Care and concern | Pricing<br>Global warming<br>Pollution<br>Waste<br>Social concerns | Positive feelings and expressions of compassion toward others and the planet. | 61 |
| Style | | Style of the clothing presented in the message. | 4 |

**Table 8.** *Cont.*

| Primary Message Themes | Message Subthemes | Description | Total Frequency |
|---|---|---|---|
| Message tone | | The general character of the message. | 44 |
| Educational | | The message uses instruction and data to increase knowledge. | 5 |
| Negative | | The message uses scare tactics, threats, or coercion, or the participant expresses doubts or reservations about brand intentions and sustainable impact. | 8 |
| Persuasive | | The message uses reason to encourage sustainable behaviors. | 1 |
| Positive | Caring for community Caring for nature Hope | The message uses optimistic substance, such as caring for society, caring the environment/nature. | 11 |
| Realistic | | The message presents realistic solutions. | 2 |
| Relevance | | The message demonstrates how sustainability relates to the consumer; consumer impacts. | 7 |
| Transparency | | Demonstration of brand transparency about practices. | 10 |
| Brand loyalty | | The participant is more concerned about the brand over the message content or tone. | 6 |
| Don't believe it | | Participant does not believe in sustainability. | 2 |
| Don't care/not important | | The participant said they did not care about practicing sustainable behaviors. | 8 |
| None | | Participant claimed that there were no memorable messages. | 6 |
| Unclear response | | Participant provided a response that was unrelated to the question. | 11 |
| Unsure | | Participant claimed that they did not know how to define it. | 12 |
| No answer | | Participant did not provide a response. | 1 |

For message content, two subthemes emerged from the data. Firstly, we noted the subtheme of *care and concern*, which implies positive feelings and expressions of compassion toward others ($n = 61$). The care and concern theme was further fragmented (i.e., environmental concerns, pricing concerns, and social concerns).

"When fashion companies show how damaging our species is to the planet".

"Making positive contributions to help save the planet was a very impactful message to me because I realized just how much each purchase truly makes a difference in the world".

"Sustainable companies are creating a safe and healthy working environment and by providing liveable wages".

The second subtheme to emerge was messages about *concerned actions and practices* ($n = 34$), which means that the company/brand message directly displayed sustainable behaviors that could be mirrored by the consumer. Specifically, these are messages about reclamation (i.e., recycling and reusing), the use of products that promise longevity, and messages about materials.

"Advertisements for organic, cruelty-free materials, post-consumer products, dedication to recycling. These remind me to be kind to the Earth".

"I try reading the labels more. What is their mission statement. What commitments they are making. Especially on a recyclable global economy".

For message tone, the most frequent theme was transparency ($n = 10$), the extent to which a business shares information about corporate practices and information. The

next most popular tone was relevance (*n* = 7), a message tone that demonstrates how sustainability relates to the consumer. Messages that used instruction and data to increase knowledge (*n* = 5) and messages that used scare tactics, threats, or coercion also resonated with the study participants (*n* = 5).

"I listen carefully to their ads and when they're transparent about what they represent".

"Messages such as what the fashion industry's effects on pollution levels and climate change are. I don't want to contribute to those effects".

"Messages including the impact I'm making by purchasing sustainable clothing make me feel validated in my decision to purchase sustainable".

"The ones that feel personal, they connect to the reader".

"I would be encouraged to purchase sustainable fashion if there were more explanations of what makes it sustainable and what the benefits of sustainable fashion are vs. regular fashion".

"I like seeing numbers and data".

## 5. Discussion

The current study focused on how we can use positive psychology to encourage change in consumers. Specifically, how can we potentially change individual behavior that could contribute to the reconstruction of social practices (e.g., reducing consumption), which could, in turn, influence other people's behavior? The importance of living sustainably, as well as the thought process behind the decisions we make and how this influences our lifestyle, offers psychologists new opportunities to promote mitigation, advance psychological understanding, and develop better solutions. The findings of the current study present several opportunities for practice and further research.

### 5.1. Consumer Demographics

Sustainability knowledge gaps between older consumers (i.e., Boomers and Gen X) and younger consumers (i.e., Millennials and Gen Z) can explain the differences in apparel purchase decisions [81–83]. Younger consumer segments are more open to lesser-known brands and driven by the authenticity of claims of environmental and social responsibility, information that is actively sought on social media [82]. However, like Zhang et al. [84], the current study demonstrated that age was not a significant factor with respect to supporting attitudes and behaviors toward sustainability. Although individuals under the age of 30 will most likely feel the long-term effects of the environmental crisis, participants in the current study over the age of 30 demonstrated similar sentiments toward sustainability, including changes in their lifestyles to lessen their environmental impact.

Gender is also a common driving factor for consumer awareness and purchase decisions, including sustainable fashion [81,85,86]. According to Gazzola et al. [81], women tend to have a stronger sense of altruism and dedication to sustainability principles in fashion. However, as with age, the current study did not reveal any gender variation with respect to consumer attitudes toward sustainability. Consumer sentiments toward sustainability are motivated or prevented by other influential, non-demographic factors, such as the COVID-19 pandemic.

### 5.2. Barriers to Sustainable Fashion Consumption

The COVID-19 pandemic has had a significant impact on society, including how people think about sustainability and fashion [3]. There has been a growing awareness of the environmental impact of the fashion industry, and the pandemic has further highlighted the need for more sustainable practices. During the peak of the pandemic, people were spending more time at home, and this led to a decrease in the consumption of fashion items, particularly fast fashion. This has provided an opportunity for individuals to reflect

on their consumption patterns and consider the impact of their actions on the environment. Consumers are interested in sustainability and being sustainable consumers, as shown in the data. However, while COVID-19 may have raised awareness, it did not result in action, indicating that barriers still exist that hamper societal-level change. The financial strain caused by pandemic-related unemployment affected the personal finances of many households [87,88] and was cited as being a significant barrier to actual sustainable consumerism. Monetary barriers have commonly inhibited sustainable behaviors in consumers who shop on a budget and perceive sustainable clothing as more expensive [89–91]; the pandemic simply exacerbated the issue.

Furthermore, a general lack of awareness and/or knowledge about sustainability is a fundamental barrier to behavior modification toward sustainable fashion consumption. Many people still do not understand the complexity that is sustainability. Some participants claimed that sustainable fashion is too costly, while others claimed to purchase more because sustainable fashion is more affordable (i.e., second-hand and vintage products). There are many opportunities for businesses to break down consumer confusion and provide clarity through transparency and altruistic messages because what they are currently doing is not enough. The need for education is also evidenced in the cynicism of participants' responses, where claims were made that the environmental crisis is a situation contrived by governments and even manufacturers for capitalist gains. Hence, while consumers' sustainable fashion behaviors as an aftermath of the pandemic cannot alone be considered adequate to foster long-term value shifts, their raised awareness can be capitalized on and complemented with businesses' sustainability-related endeavors to achieve the same. In other words, our results indicate a need for businesses and consumers to develop a meaningful partnership, bringing their own pandemic-resulting experiences to create long-term sustainable fashion behaviors in consumers.

Participant consumers reported their concerns about social and environmental causes on several occasions, and such concerns were amplified due to the industry-wide sustainability awareness as identified by prior research. However, a lack of resulting action, along with their references to government policies and businesses' motives, indicates that while they intend to be sustainable, they do not presume any responsibility to execute it. Similarly, while participants reported their care, altruism, and affect (along with product characteristics and self-focused reasons) as being inspirational regarding their sustainable fashion purchases pre- and post-pandemic and acknowledged their positive emotional outcomes as a result of such transactions, they resorted to the businesses' marketing messages for information that evoked such positive antecedents and consequences, to begin with. Accordingly, one may conclude that while positive psychology can potentially be a major motivator for consumers to adopt sustainable consumption practices, businesses need to be the patrons for action and education to pave the path for long-term sustainability.

## 6. Conclusions

### 6.1. Major Findings

This study's aims were twofold: (1) to examine the impact of the pandemic on long-term sustainable fashion consumer behavior and (2) to investigate the effectiveness of marketing strategies in promoting long-term sustainable consumption behavior. Through the summative content analysis and thematic analysis of data collected from 154 online survey participants, our findings indicated that consumers' attitudes and behaviors toward sustainable fashion did not significantly change during the pandemic, contrary to prior literature's claims. However, the study revealed that consumers still face challenges in understanding what sustainability entails and identified cost and lack of awareness as the primary barriers to sustainable fashion consumption post-pandemic.

However, our analysis also demonstrated that consumers' pro-sustainability tendencies were influenced not only by positive emotions associated with care and concerns but also by positive emotions experienced after making a purchase, such as contentment and altruism. These results emphasize the importance of incorporating positive psychology per-

spectives into marketing messages to motivate long-term sustainable fashion consumption practices effectively.

### 6.2. Theoretical Implications

Our study contributes to the existing literature on sustainable fashion consumption by deepening our understanding of consumer motivations and providing valuable implications for post-pandemic sustainable fashion consumption. Our findings highlight three key points.

Firstly, we found that environmental concerns are the most significant motivation for sustainable fashion consumption, which was consistent before and after the pandemic. Environmental concern (EC) is a value-based attitude toward environmental issues [92]. Consumers' long-term value for environmental issues is reflected in the consistency of their environmental concerns. Research shows that consumer values play a significant role in crises such as the pandemic [49,50], and consumers tend to be loyal to products and brands that match their values [51]. Therefore, environmental concerns are likely to remain a value-oriented attitude and hold longevity in consumers' shift post-pandemic.

Secondly, we found that, post-pandemic, participants were more interested in anything related to them. Therefore, they paid more attention to advertising and marketing that deliberately sought their attention. Our data showed that participants were motivated by their own interests and resources, including financial resources, to purchase sustainable goods. The sustainable fashion market can be clustered by consumers' diverse interests and resources.

Thirdly, our findings showed that the participants were motivated by positivity. For both pre- and post-pandemic, they experienced positive emotions as a result of purchasing sustainable fashion. Marketing messages that resonated with participants were primarily those with positive or altruistic content and tone. These positive emotions may underlie long-term sustainable consumption behavior as emotional experience and benefits identified by consumers are strong determinants of loyalty to products and brands [93], suggesting a long-lasting determinant of behavior.

### 6.3. Practical Implications

The findings of this exploratory study further provide practical implications for businesses to foster and promote pro-sustainability consumerism. As the study started by asking whether the COVID-19 pandemic affected consumers and their attitudes and behaviors in favor of sustainable fashion, important points to be emphasized in this context are the subjective meaning of sustainability and its role in considering sustainable fashion, and how these were affected by the COVID-19 pandemic.

First, the study's findings revealed an uneven understanding of sustainability among consumers, primarily associating it with resolving environmental concerns. Participants also reported their overwhelming reliance on brands' marketing messages regarding being inspired and practicing sustainable fashion consumption. Consequently, businesses must prioritize consumer education to address this knowledge gap about the complexity of sustainability. Specifically for the fashion industry, as businesses spend millions of dollars to commit to various sustainable practices [94], educating their consumers becomes essential to garner a loyal customer market. These findings corroborate with prior research conducted before the pandemic [34,90,95], indicating a persistent need for sustainability knowledge among fashion consumers. Despite the pandemic's supposed pivotal role in the industry's sustainability journey [96], our results suggest that the lack of accurate knowledge of sustainable fashion continued to cloud consumers' thoughts both pre- and post-pandemic. Accordingly, businesses should focus on educating consumers about the holistic nature of sustainability, encompassing societal, economic, and environmental aspects, given that fashion brands commit to all three as core business strategies.

Second, our findings emphasize the importance of using marketing messages to cultivate empathy in fashion consumers, promoting more sustainable consumption behavior.

Results indicated that most participants' sustainable fashion consumption remained unchanged despite COVID-19 and they cited a general lack of awareness and concern as common barriers in pre- and post-pandemic sustainable fashion consumption. Accordingly, fashion brands are encouraged to not only educate consumers but also instill a sense of care and compassion for all stakeholders within the fashion industry. During global crises like the COVID-19 pandemic, empathetic marketing messages can guide consumers toward more sustainable fashion choices, considering their reliance on message content for consumption decisions. By appealing to consumers' altruism, marketing can play a pivotal role in fostering sustainable fashion consumption during such challenging times.

Third, this study recommends highlighting positive emotions in fashion businesses' marketing messages to promote sustainable fashion consumption. As the study findings indicated, consumers who made sustainable fashion purchases before and after the pandemic reported positive feelings, including feeling contented, joyous, confident, altruistic, and happy. Accordingly, fashion brands should associate sustainable fashion with positive emotions to encourage consumers to engage in sustainable consumption. Such strategies are already acknowledged by researchers and implemented by the tourism industry to foster pre-sustainable behaviors [97]. We recommend the same for fashion brands, focusing on and highlighting emotional benefits in their marketing messages to promote sustainable consumer behaviors.

## 7. Limitations and Future Research

This study had limitations that warrant discussion. Our data revealed that a significant number of respondents either provided no answer or gave unclear responses when asked about the concept of sustainability and sustainable fashion. This gap in understanding is overlooked in the existing literature on sustainability, and clarifying these concepts is necessary to study consumers' sustainability-related behaviors and values. Therefore, future research should consider segmenting consumers by their level of understanding of sustainability to gain more insights into sustainable consumption. Such segmentation would provide marketers with valuable information on their target consumer groups.

It is important to note that economic conditions and status played a crucial role in shaping consumers' reactions to the pandemic. Financial resources influence consumers' risk management ability [59] and their options for sustainable fashion, such as choosing sustainable fashion brands or buying second-hand clothing. However, our study did not collect respondents' financial information, including their household or individual income levels, and only gathered limited demographic data. Although large and diverse, our sample does not represent the experiences of all individuals during and after the COVID-19 pandemic. Therefore, future research should analyze sustainable consumer behavior in relation to consumers' financial resources to gain a better understanding of the determinants of consumption behavior.

Moreover, while the study findings provided some important insights concerning the consumers' sustainable fashion consumption behavior and how it was affected by the COVID-19 pandemic to promote long-term changes, given the small sample size, these findings cannot be generalized. Furthermore, this research focused only on US participants to understand the effect of the pandemic on consumer sustainable fashion consumption behavior. As the pandemic extended globally, consumers across different countries are expected to be affected differently. Therefore, future research is recommended to include a larger sample from the US and other countries, to improve the generalizability and present a comparative understanding of the findings.

Finally, all data were based on self-reporting, and the participants may have underreported their attitudes and behaviors on certain questions. Thus, our findings may underestimate the impact of the COVID-19 pandemic on participants' consumption behaviors. Future research could employ a more theoretically driven framework that explores quantifiable variable relationships.

**Author Contributions:** Conceptualization, J.S. and S.G.; Funding Acquisition, J.S., S.G. and J.H.K.; Investigation, J.S.; Data Curation, J.S. and R.L.; Methodology, J.S., S.G. and J.H.K.; Project Administration, J.S.; Resources, J.S.; Visualization, J.S., S.G., J.H.K. and R.L.; Validation, J.S. and R.L.; Writing—Original Draft, J.S., S.G. and J.H.K.; Writing—Review and Editing, J.S., S.G., J.H.K. and R.L. All authors have read and agreed to the published version of the manuscript.

**Funding:** This research was funded by the University of Rhode Island Social Science Institute for Research, Education, and Policy.

**Institutional Review Board Statement:** The study was conducted in accordance with the Declaration of Helsinki and approved by the Institutional Review Board of the University of Rhode Island (IRB2122-030 on 24 August 2021).

**Informed Consent Statement:** Informed consent was obtained from all subjects involved in the study.

**Data Availability Statement:** The complete datasets can be requested from the first author.

**Conflicts of Interest:** The authors declare no conflict of interest.

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
