# Peer review of "Improving Society and the Planet: Sustainability and Fashion Post-Pandemic"

_sustainability, doi:10.3390/su151712846_

Round 1
Reviewer 1 Report
Dear Authors,
Many thanks for the opportunity to read your manuscript. There are some ideas for improvement of your manuscript that you will need to address:
The gap is not clearly identified and still unclear do we really need and have the lack of such kind research. As the introduction is already too long maybe it is better to work on it decrease and add the aim and some basic facts on methodology.
In 2.1 authors it has provided a high-quality literature review but during the whole manuscript no mentions about sustainability marketing. It will be good to mention this point in such a framework. There is good work for enlightenment: Burksiene, V., Dvorak, J., & Burbulyte-Tsiskarishvili, G. (2018). Sustainability and sustainability marketing in competing for the title of European Capital of Culture. Organizacija, 51(1), 66-78.
In 2.2 you mentioned a healthy and caring economy. It will be good to describe your definition of it.
In 2.2 you mentioned that new habits exist. Can you give some examples?
According to the MDPI style it must not be like this: Phipps and Ozanne [63] demonstrated that individuals
2.4 In 2.4 unclear what is the project aim is mentioned in the manuscript.
RQ 2 and 3 are questions Yes/NO
3.1 what was planned for the initial number of respondents? How it happened that you were not able to mobilize more respondents?
Maybe you can remove limitations to the introduction and instead, them have normal conclusions with future research.
All the best
Author Response
Thank you for your feedback. we did our best to address every comment and hope our responses have strengthened the manuscript.
Our responses to your comments are in red.
____
REVIEWER 1
Many thanks for the opportunity to read your manuscript. There are some ideas for improvement of your manuscript that you will need to address:
The gap is not clearly identified and still unclear do we really need and have the lack of such kind research. As the introduction is already too long maybe it is better to work on it decrease and add the aim and some basic facts on methodology.
- We have rephrased and shortened the Introduction to clearly articulate the research gaps. We have also included a brief description of the methodology and made the aim of the study more apparent.
In 2.1 authors it has provided a high-quality literature review but during the whole manuscript no mentions about sustainability marketing. It will be good to mention this point in such a framework. There is good work for enlightenment: Burksiene, V., Dvorak, J., & Burbulyte-Tsiskarishvili, G. (2018). Sustainability and sustainability marketing in competing for the title of European Capital of Culture. Organizacija, 51(1), 66-78.
- Thanks for this suggestion. However, please note that since the manuscript already refers to sustainability marketing literature (e.g., 16, 29, 32), we did not include any additional new references.
In 2.2 you mentioned a healthy and caring economy. It will be good to describe your definition of it.
- The terms were changed to ‘sustainable’ to clarify the concept on page 4
In 2.2 you mentioned that new habits exist. Can you give some examples?
- For clarity, we added “of thrifting” to the sentence as stated in the previous sentence in the paragraph.
According to the MDPI style it must not be like this: Phipps and Ozanne [63] demonstrated that individuals
- The format was updated following the MDPI style on page 5.
2.4 In 2.4 unclear what is the project aim is mentioned in the manuscript.
- The section clarifying the purpose of the project has been updated with additional information on page 13.
RQ 2 and 3 are questions Yes/NO
- Indeed, the RQs 2 and 3 are framed in the affirmative or negative format. However, this format allows for an elaborated insight on the role and efficiency of marketing to educate consumers for long-term sustainability – the primary focus of our second research objective. We would like to draw the reviewer‘s kind attention to the discussion section. Following this above-mentioned structure, the yes/no responses of these two RQs have been explained and accordingly it has been concluded that marketing indeed is vital in educating consumers to transition to a long-term sustainable consumption. As such, we believe the RQ 2 and 3 are justifiably formatted to achieve the research‘s second purpose and hope the reviewer would agree with us.
3.1 what was planned for the initial number of respondents? How it happened that you were not able to mobilize more respondents?
- This study panel was funded by a grant. Due to limited funds, we were not able to “mobilize more respondents” (i.e., lack of participant incentives). We negotiated an agreement (i.e., contract) with Qualtrics to identify potential participants based on our budget.
3.2. Maybe you can remove limitations to the introduction and instead, them have normal conclusions with future research.
- Thanks for the suggestion. We did not initially include a Conclusion section since the journal layout indicated it as optional. However, following your suggestion, we created a new Conclusion section and moved the theoretical and practical contributions from Discussion under this new section. We did not move the limitations to the Introduction since these lead to the future research needs organically without us being redundant. We hope the reviewer would agree with this reorganization.
Reviewer 2 Report
This paper is quite interesting. However, the authors are recommended to amend the paper based on the following comments:
Introduction: the authors are recommended to justify the importance of this study, and pandemic, based on statistics. The statistics should be retrieved from statista.
In addition, the authors are also recommended to discuss the importance of social media contents during the pandemic. Please read and cite the following papers:
Cheung, M. L., Leung, W. K., Aw, E. C. X., & Koay, K. Y. (2022). “I follow what you post!”: The role of social media influencers’ content characteristics in consumers' online brand-related activities (COBRAs). Journal of Retailing and Consumer Services, 66, 102940.
Ao, L., Bansal, R., Pruthi, N., & Khaskheli, M. B. (2023). Impact of Social Media Influencers on Customer Engagement and Purchase Intention: A Meta-Analysis. Sustainability, 15(3), 2744.
The authors are also recommended to present the importance and contributions of this paper briefly in the introduction section.
minor english editing is needed
Author Response
Thank you for your feedback. we did our best to address every comment and hope our responses have strengthened the manuscript.
Our responses to your comments are in red.
___
REVIEWER 2
This paper is quite interesting. However, the authors are recommended to amend the paper based on the following comments:
Introduction: the authors are recommended to justify the importance of this study, and pandemic, based on statistics. The statistics should be retrieved from statista.
- Thanks for your recommendation. While Statista is a great resource, we politely disagree with the reviewer on the need of statistics to justify the importance of this study. We have revised the Introduction to clearly articulate the research gaps and accordingly to justify the need of this study. These research gaps are supported with most current academic literature and market studies. Accordingly, we trust that the cited literature adequately supports the merit and relevance of our research and we hope the reviewer would agree with us.
In addition, the authors are also recommended to discuss the importance of social media contents during the pandemic. Please read and cite the following papers:
Cheung, M. L., Leung, W. K., Aw, E. C. X., & Koay, K. Y. (2022). “I follow what you post!”: The role of social media influencers’ content characteristics in consumers' online brand-related activities (COBRAs). Journal of Retailing and Consumer Services, 66, 102940.
Ao, L., Bansal, R., Pruthi, N., & Khaskheli, M. B. (2023). Impact of Social Media Influencers on Customer Engagement and Purchase Intention: A Meta-Analysis. Sustainability, 15(3), 2744.
- Thank you for recommending specific references and providing suggestions. We carefully reviewed the references you suggested and found that they investigate the influence of social media on consumer behavior. While these topics and findings contribute to the understanding of the impact of new media on consumer behavior, they may not be directly relevant to supporting our research questions. Our research employs qualitative approaches to explore our research questions addressing extant literature’s inconsistent findings and lack of investigation on sustainable fashion consumption regarding the pandemic,, with a particular emphasis on sustainable fashion consumer behavior rather than the specific impact of media and general consumer behavior.. Therefore, regrettably, we have decided not to incorporate the recommended references in this revision. We kindly request your understanding regarding our rationale for this decision. Thank you for your understanding.
The authors are also recommended to present the importance and contributions of this paper briefly in the introduction section
- The contributions of the current research are discussed in the last paragraph of the Introduction (page 2 and 3)
Reviewer 3 Report
Comment 1: While the research gaps are clearly outlined, it would be helpful to explicitly state why these gaps are significant and what their implications are for the field. This will further emphasize the importance of addressing these gaps and provide a stronger rationale for conducting the study.
Comment 2: There are instances of convoluted sentence structures and repetitions in the introduction. Simplify the language and revise the sentences to improve clarity and coherence. Additionally, check for grammatical errors and inconsistencies in the writing. A thorough proofreading will enhance the overall quality of the manuscript.
comment 3: Limited discussion and the article acknowledges a gap in research regarding consumers' sustainable consumption practices during the pandemic, it fails to delve deeply into the topic. It would be beneficial to explore the potential factors influencing consumers' behavior, their attitudes toward sustainability, and any observed changes during the pandemic.
Comment 4: The authors mention using an open coding stage to review the responses and derive themes from the data by identifying patterns, commonalities, and differences. However, the description of the process could benefit from more clarity and specific examples to illustrate how the themes were derived.
comment 5: Another criticism pertains to the generalizability of the study's findings. While the authors acknowledge that the sample consisted of participants from the US, they do not sufficiently justify why this sample was chosen or discuss potential implications for the generalizability of the findings to other populations or cultural contexts. The study could have provided a more comprehensive discussion on the limitations of a US-centric sample and the need for cross-cultural comparisons to capture the diversity of experiences and perspectives related to sustainable fashion consumption during the pandemic.
Minor editing of the English language required
Author Response
Thank you for your feedback. we did our best to address every comment and hope our responses have strengthened the manuscript.
Our responses to your comments are in red.
_________________________
REVIEWER 3
Comment 1: While the research gaps are clearly outlined, it would be helpful to explicitly state why these gaps are significant and what their implications are for the field. This will further emphasize the importance of addressing these gaps and provide a stronger rationale for conducting the study.
- We have explained these on page 1 and 2 in the last paragraph of the introduction.
Comment 2: There are instances of convoluted sentence structures and repetitions in the introduction. Simplify the language and revise the sentences to improve clarity and coherence. Additionally, check for grammatical errors and inconsistencies in the writing. A thorough proofreading will enhance the overall quality of the manuscript.
- The authors have proofread the manuscript and corrected all grammatical errors.
comment 3: Limited discussion and the article acknowledges a gap in research regarding consumers' sustainable consumption practices during the pandemic, it fails to delve deeply into the topic. It would be beneficial to explore the potential factors influencing consumers' behavior, their attitudes toward sustainability, and any observed cshanges during the pandemic.
- Thank you for your comment. It seems our findings were not clearly examined in the discussion of the previous version of the manuscript. To address this, we have made some changes in our revision. We created a separate conclusion section where we summarize the major findings, including the observed changes in sustainable fashion behavior and attitudes during the pandemic, as well as potential determinants of long-term shifts in consumer sustainable fashion behavior. Additionally, we have repositioned the academic and practical implications section right after the conclusion to highlight the findings and their implications. We hope these revisions clarify your concerns and improve the overall clarity of our manuscript.
Comment 4: The authors mention using an open coding stage to review the responses and derive themes from the data by identifying patterns, commonalities, and differences. However, the description of the process could benefit from more clarity and specific examples to illustrate how the themes were derived.
- This is explained in detail the Methods section on page 6-7:
“The codebook continued to develop through the inductive process [72, 74] by looking for the presence of attributes and the frequency of occurrence of key words and ideas as we coded the dataset, and then identified patterns (i.e., commonalities and differences) to derive themes from the data [76, 77].”
“During the second phase of coding, the themes were checked against the coded data. In this stage, several categories were eliminated, collapsed, and combined into central themes based on the similarities in responses. The broader categories were systematically developed into subcategories. During the third, and final, phase of coding the first author reviewed and refined the themes after cross checking the categories with the terms and phrases used in the sustainability and consumer psychology literature [e.g., 68, 78]”
We used an inductive process, which is outlined step-by-step. A few supplementary sentences have been added for readers who do not understand inductive analysis (on page 6).
comment 5: Another criticism pertains to the generalizability of the study's findings. While the authors acknowledge that the sample consisted of participants from the US, they do not sufficiently justify why this sample was chosen or discuss potential implications for the generalizability of the findings to other populations or cultural contexts. The study could have provided a more comprehensive discussion on the limitations of a US-centric sample and the need for cross-cultural comparisons to capture the diversity of experiences and perspectives related to sustainable fashion consumption during the pandemic.
- We explain in the methods section (see page 6) that we used a market research service. We were limited (financially) in the number of participants we could recruit. Therefore, we decided that it was best to limit the geographic scope to the US so that we could make adequate assumptions/interpretations of the data.
Reviewer 4 Report
Thanks for the opportunity to review the article.
1. I find this article interesting, written well but still needs more elaboration and clarification. Authors have written very important lines “it is not just the fashion brands and their market environments; COVID-19 drastically impacted consumer behaviors as well [3]. For example, scholars reported that with the government-imposed lockdowns, traditional consumption of branded fashion
products was replaced with surges in domestic consumptions, such as repairs and do-it yourself consumptions” which I believe it is the core of the introduction section. Nevertheless, I still want to see how the term sustainability is linked with the fashion brands , does it mean that Covid19 has forced customers to save and reuse their available clothes and accordingly this is considered a sustainability. Please clarify it more as this is very important, readers should be able to easily understand the meaning.
2. I have checked the article and I feel like there are some grammar mistakes and long sentences which need to be shortened.
3. Please include conclusion section for the article.
4. Please discuss the implications more comprehensively.
5. please try to shorten some sections pf the analysis and combine them with other sections. so many details are there , it becomes confusing.
all the best
NA
Author Response
Thank you for your feedback. we did our best to address every comment and hope our responses have strengthened the manuscript.
Our responses to your comments are in red.
_______________________
REVIEWER 4
- I find this article interesting, written well but still needs more elaboration and clarification. Authors have written very important lines “it is not just the fashion brands and their market environments; COVID-19 drastically impacted consumer behaviors as well [3]. For example, scholars reported that with the government-imposed lockdowns, traditional consumption of branded fashion products was replaced with surges in domestic consumptions, such as repairs and do-it yourself consumptions” which I believe it is the core of the introduction section. Nevertheless, I still want to see how the term sustainability is linked with the fashion brands , does it mean that Covid19 has forced customers to save and reuse their available clothes and accordingly this is considered a sustainability. Please clarify it more as this is very important, readers should be able to easily understand the meaning.
- This is a very good question. To clarify, prior to the pandemic, fashion brands addressed sustainability by tackling social and environment issues, as mentioned in the second paragraph of Literature Review 2.1 (on page 3). Similarly, from the consumers’ standpoint, some scholars regarded practices involving less disposal and more reuse as a way to practice sustainability. Hence, this study accordingly assumed and labeled all such practices as sustainable ones. However, since we were interested in exploring if and how these practices transformed or evolved during the pandemic, we intentionally refrained from presenting a specific working definition to the participants. This approach allowed us to capture our participants’ true and unadulterated comprehension of sustainable fashion. We have added a brief explanation of this approach in the Method section 3.2.
- I have checked the article and I feel like there are some grammar mistakes and long sentences which need to be shortened.
- The authors have proofread the manuscript and corrected all grammatical errors.
- Please include conclusion section for the article.
- Thanks for the suggestion. Following your suggestion, we created a conclusion section and a summary of major findings were added to the section. Additionally, we moved the theoretical and practical implications to the conclusion section from Discussion under this new section.
- Please discuss the implications more comprehensively.
- We have revised both theoretical and practical implications, making them more succinct.
- please try to shorten some sections pf the analysis and combine them with other sections. so many details are there , it becomes confusing.
- True to the nature of qualitative inquiry, the findings section will be the longest section of the manuscript. We have tried to consolidate the nuances of coding (e.g., codes and definitions) into comprehensive tables to save text space. Findings were shortened to the extent that it does not limit an understanding of the data. Also, the tables are included to provide clarity for reader confusion. If the editors would like, the tables (all or some) can be submitted as supplementary information.
Reviewer 5 Report
Dear Authors
It seems that you have put great effort into this study. As the study was funded, you have followed all the necessary steps.
1. The study seems almost perfect but is a bit lengthy. If possible, cut down the first two sections, as they should not be detailed this much.
2. Major concerns: too many references. Many are too old and obsolete. It must be cut down. Keep only the original an latest relevant citations.
3. Study is okay. well done.
Author Response
Thank you for your feedback. we did our best to address every comment and hope our responses have strengthened the manuscript.
Our responses to your comments are in red.
______________________
REVIEWER 5
It seems that you have put great effort into this study. As the study was funded, you have followed all the necessary steps.
- The study seems almost perfect but is a bit lengthy. If possible, cut down the first two sections, as they should not be detailed this much.
- We have shortened the Introduction.
- Major concerns: too many references. Many are too old and obsolete. It must be cut down. Keep only the original an latest relevant citations.
- After reviewing our references, we have determined that the majority of the references, except for two, are relevant to the thorough discussion of our topic and subsequent findings in relation to the existing literature. Accordingly, two references have been deleted.
- Study is okay. well done.
- Thank you.
Round 2
Reviewer 1 Report
Dear Authors,
Thank you for the updated manuscript. I think it can be published in a Sustainability journal.
All the best
Author Response
Thank you.